# Determination of Volatile Organic Compounds in Water by Attenuated Total Reflection Infrared Spectroscopy and Diamond-Like Carbon Coated Silicon Wafers

**Carina Dettenrieder [1], Dervis Türkmen [1], Andreas Mattsson [2], Lars Österlund [2]** 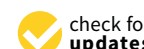**, Mikael Karlsson [2],* and Boris Mizaikoff [1],***

[1] Institute of Analytical and Bioanalytical Chemistry, Ulm University, Albert-Einstein-Allee 11, 89081 Ulm, Germany; carina.dettenrieder@uni-ulm.de (C.D.); dervis.tuerkmen@uni-ulm.de (D.T.)

[2] Department of Materials Science and Engineering, The Ångström Laboratory, Uppsala University, P.O. Box 534, 75121 Uppsala, Sweden; andreas.mattsson@angstrom.uu.se (A.M.); lars.osterlund@angstrom.uu.se (L.Ö.)

* Correspondence: mikael.karlsson@angstrom.uu.se (M.K.); boris.mizaikoff@uni-ulm.de (B.M.); Tel.: +46-18-471-7237 (M.K.); +49-731-50-22750 (B.M.)

**Abstract:** Volatile organic compounds (VOCs) are one of the most commonly detected contaminants in water. The occurrence is mainly in gasoline and other petroleum-based products, fumigants, paints and plastics. Releases into the environment and the widespread use have an impact on the ecosystem such as humans and animals due to their toxicity, mutagenicity, and carcinogenicity. VOCs may persist in groundwater and may enter drinking water supplies. In this paper, a diamond-like carbon (DLC)-coated silicon waveguide in combination with a polymer film (ethylene/propylene copolymer, E/P-co) for enrichment of analytes was investigated to determine its suitability for ATR-FTIR (attenuated total reflection Fourier transform infrared) spectroscopic detection of VOCs. The DLC film was fluorine-terminated enhancing the adhesion of the hydrophobic polymer to the waveguide surface. The analytes diffuse into the hydrophobic polymer whereas water is excluded from the emanating evanescent field. Therefore, direct detection in aqueous systems is enabled. Nine VOCs, i.e., ethylbenzene (EB), trichloroethylene (TCE), tetrachloroethylene (TeCE), the xylene isomers (*p*-xylene, *p*XYL; *m*-xylene, *m*XYL; *o*-xylene, *o*XYL), naphthalene (NAPH), toluene (TOL), and benzene (BENZ), were evaluated simultaneously qualitatively and quantitatively showing the potential of DLC coatings revealing high sensitivities in the low ppb to ppm concentration range, i.e., 50 ppb for TeCE. To the best of our knowledge, this is the first time of IR spectroscopic detection of VOCs in aqueous solutions using DLC-coated waveguides in combination with a hydrophobic polymer. By utilizing a DLC-coated waveguide, a versatile sensor for real-time monitoring in harsh environments such as effluents, leaking pipelines, and underground storage tanks is feasible due to response times within a few minutes.

**Keywords:** mid-infrared sensor; volatile organic compounds; attenuated total reflection spectroscopy; diamond-like carbon; hydrophobic polymer coating; ethylene/propylene copolymer

---

## 1. Introduction

Volatile organic compounds (VOCs) are contained in gasoline, diesel fuel and various petroleum-based products and originate from industrial effluents, sewage disposal, oil storage wastes, and oil tanker accidents. Many of them are highly toxic, mutagenic, and carcinogenic.

Due to their production and widespread use, VOCs are one of the most commonly detected organic pollutants in water. Releases into the environment may occur resulting in damages to humans and the environment [1–5].

Monitoring of pollutants, particularly VOCs, in global waters including seawater, river and lake water, ground, surface and drinking water is important to maintain public health and protect the environment. The most commonly analytical techniques for detection of VOCs are chromatographic methods, i.e., gas chromatography and high-performance liquid chromatography [3,6–9]. However, these methods include bulky measurement equipment and are therefore, confined to laboratory use. Furthermore, trained personnel are needed due to complex sample pretreatment steps. Therefore, these methods are very time-and cost-intensive. Hence, these techniques are not appropriate for monitoring in real-time and on-site.

Sensors based on ATR-FTIR spectroscopy are emerging one of the most suitable methods for detection of pollutants. Specific vibrations (i.e., stretching, bending and rotating) of the molecules in the "fingerprint" region (1500–500 cm$^{-1}$), enable direct qualitative analysis to distinguish different analytes [10]. These sensors offer continuous on-site and in real-time monitoring of pollutants. Therefore, the sample does not have to be transported to the laboratory. Hence, falsified results due to the volatility of the analytes and point-sampling are prevented, time is saved, and costs are reduced.

In ATR-FTIR spectroscopy, the IR beam is totally internally reflected and an evanescent field at every reflection is established. According to Snell's law, the incident angle has to be larger than the critical angle $\theta_c$ as described in Equation (1) with the refractive indices of the waveguide $n_1$ and surrounding medium, i.e., polymer membrane, $n_2$.

$$\theta_c = \arcsin\left(\frac{n_2}{n_1}\right) \tag{1}$$

Detection directly in aqueous systems is enabled due to hydrophobic polymer-coated waveguides. Hydrophobic analyte molecules (i.e., VOCs) are enriched, whereas water is excluded from the penetration depth of the evanescent field. Therefore, high interfering water absorptions are prevented. The penetration depth $d_p$ can be calculated as follows:

$$d_p = \frac{\lambda}{2\pi \sqrt{n_1^2 \sin^2 \theta - n_2^2}} \tag{2}$$

With the incident wavelength $\lambda$, the angle of incidence $\theta$ at the interface between waveguide and the surrounding medium (i.e., polymer membrane). The evanescent field intensity is dependent on the distance to the surface waveguide $x$ and the penetration depth [11].

$$E(x) = E_0 \exp\left(-\frac{x}{d_p}\right) \tag{3}$$

ATR-FTIR waveguide materials include crystals, e.g., zinc selenide (ZnSe), zinc sulfide (ZnS), silicon (Si), and germanium (Ge) and thin film waveguides based on gallium arsenide/aluminum gallium arsenide (GaAs/AlGaAs), mercury-cadmium-telluride (MCT), and diamond [12–15]. Alternatively, optical fiber waveguide materials were already reported, e.g., chalcogenide, silver and tellurium halide [10,16–22]. In particular, silver halide fibers are one of the most promising materials due to their flexibility and transparence in the entire mid-infrared (MIR) spectral regime [23]. However, silver halide fibers decompose upon contact with seawater due to the high amount of chloride ions, furthermore, silver halide fibers are susceptible to ultraviolet radiation. Since a sensor withstanding in difficult matrices, i.e., seawater with its high ionic strength and high content of organic matter, is envisaged, the optical fiber has to be resistant to ever-changing conditions. Hence, a protective coating of the silver halide fiber is strictly necessary.

In order to detect VOCs in aqueous solutions, the waveguide has to be coated with a hydrophobic polymer membrane. The hydrophobic pollutants diffuse into the polymer layer, whereas the water matrix is excluded. At the same time, this polymer layer acts as protective coating of the silver halide fiber [24–26]. Various types of hydrophobic polymer membranes were already investigated, e.g., low-density polyethylene (LDPE) [17,27,28], Teflon®AF [29–31], poly(dimethylsiloxane) (PDMS) [32–34], polyisobutylene (PIB) [17,35–37], and E/P-co polymer [16,35,38–41]. However, within harsh environments the polymer coating can come off. Hence, sufficient adhesion of the polymer to the waveguide surface has to be ensured.

Films of DLC are composed of a $sp^3$-and $sp^2$-hybridized carbon. Hence, DLC offers a high robustness and chemical inertness with transparency in the entire IR spectral range [42–44]. Janotta et al. have successfully demonstrated the feasibility of DLC films as protective coating for detecting strongly oxidizing agents in aqueous matrix [45]. DLC films have the advantage of having an amorphous structure offering the possibility of surface termination with e.g., hydrogen, oxygen, or fluorine tailoring the features of the waveguide surface [46]. Therefore, a fluorine-termination reveals a highly hydrophobic surface offering a strong connection to the applied hydrophobic polymer coating. DLC films would enhance the long-term durability hindering the decomposition of the silver halide fiber, and therefore, preventing the attenuation of the transmitted IR radiation [47]. DLC in combination with MIR sensing was already presented by several research teams [43,45,48–50], and application within changing conditions, i.e., high temperature and humidity, was already shown [51]. Therefore, application in harsh environments and sufficient protection of the waveguide is enabled. To our knowledge, there are no literature reports on the use of DLC-coated waveguides combined with hydrophobic polymers for IR spectroscopic detection of VOCs in water.

In this study, the potential of F-terminated DLC-coated Si wafers in combination with a hydrophobic E/P-co polymer layer for pollution monitoring, i.e., direct detection of VOCs, via ATR-FTIR spectroscopy is demonstrated. The fluorine surface termination of the DLC layer enables strong binding to the polymer membrane. Therefore, water molecules are efficiently rejected from the evanescent field while analyte molecules diffuse inside the membrane. Nine VOCs could be simultaneously detected providing detection limits in the ppb to ppm concentration range. Multiple regenerations with an aqueous methanol solution did not influence the adhesion of the polymer layer to the DLC-coated waveguide. The sensor responds very fast due to direct change in peak area of the respective evaluated analyte peak, i.e., using the kinetic method by evaluating the slope of the tangent, qualitative results are obtained within the first minutes after starting the measurement. After a short period of 5 to 10 min partition equilibrium has been achieved and evaluation can be performed via equilibrium method for more precise results, This is very important for a fast detection of leaks, i.e., underground fuel storage tanks and pipelines, and real-time monitoring of occurring VOCs in wastewater in order to protect human and animal health and the environment.

## 2. Materials and Methods

### 2.1. Chemicals and Reagents

TCE (≥99.5%, CAS number: 79-01-6), TeCE (≥99.9%, CAS number: 127-18-4), *p*XYL (≥99.0%, CAS number: 106-42-3), *m*XYL (≥99%, CAS number: 108-38-3), *o*XYL (≥98.0%, CAS number: 95-47-6), NAPH (≥99.0%, CAS number: 91-20-3), EB (≥99.0%, CAS number: 104-41-4), TOL (≥99.8%, CAS number: 108-88-3), BENZ (≥99.9%, CAS number: 71-43-2) and methanol (MeOH, ≥99.8%, CAS number: 67-56-1) were purchased from Sigma-Aldrich Sweden AB (Stockholm, Sweden). E/P-co polymer (60:40) (CAS number: 9010-79-1) was obtained from Aldrich Chemical Company (Milwaukee, WI, USA). All chemicals were used without further purification steps.

## 2.2. Sample Preparation

Si wafers were cleaned using standard RCA 1 and 2 procedure. Subsequently, the wafers were dipped into aqueous HF solution (1:50). The cleaned Si wafers were treated with a pulsed filtered cathodic arc deposition with a high-purity graphite cathode in a deposition chamber (base pressure: $10^{-5}$ Pa) obtaining a DLC layer (tetragonal amorphous carbon, ta-C) with a thickness of 30 nm. At the end of the DLC deposition step, $SF_6$ was induced into the deposition chamber (pressure 1 Pa) for surface termination of the DLC revealing a highly hydrophobic surface (F:a-C).

Subsequently, the F-DLC-coated Si wafer was coated with E/P-co polymer. Therefore, solid E/P-co was dissolved under reflux in *n*-hexane resulting in a 1% E/P-co solution. 20 µL of the solution were dip-coated using an Eppendorf pipette. After evaporation of the solvent a thin film of E/P-co with a thickness of 8.23 ± 0.22 µm, determined via differential weighing [16], was obtained. A schematic of the F-DLC E/P-co polymer coated Si wafer is illustrated in Figure 1. Prior to measurement, the polymer membrane was exposed to water for at least 24 h for polymer conditioning.

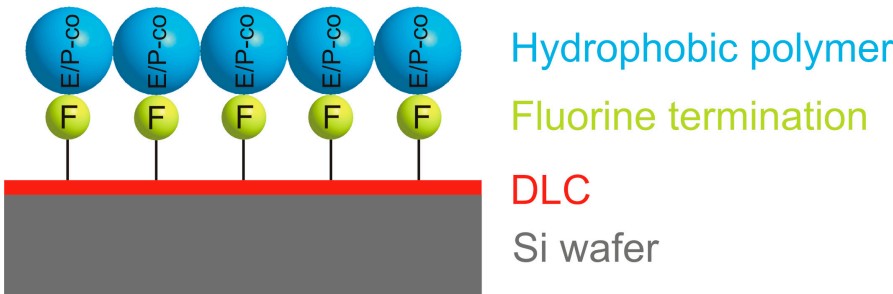

**Figure 1.** Schematic diagram of final chip structure. Fluorine terminated DLC was applied to the Si wafer surface. Subsequently, the hydrophobic F-DLC surface was coated with E/P-co polymer membrane for enrichment of hydrophobic analyte molecules.

Since the analytes are highly hydrophobic, and therefore, poorly soluble in water, stock solutions in MeOH with a concentration of 2000 ppm each, were prepared in 20 mL headspace vials. A specific amount of the stock solution was dissolved in deionized water obtaining the respective concentration. MeOH acts simultaneously as a solubility mediator, but do not influence the measurement performance [51]. The concentration of MeOH was kept at 1% (*v/v*).

## 2.3. Instrumentation and Data Acquisition

ATR-FTIR spectroscopic measurements were performed with a commercial vacuum-pumped Bruker IFS 66v/S FTIR spectrometer (Bruker Optics, Ettlingen, Germany) in combination with a BioATR II cell and a liquid nitrogen-cooled MCT detector. The BioATR II cell was equipped with an E/P-co polymer coated F-DLC-coated Si ATR chip with 13 internal reflections. The analyte solution containing the nine analytes was flushed through the cell using a peristaltic pump (GE Healthcare, Chicago, IL, USA). The flow rate was set to 6 mL/min. IR spectra were recorded in the MIR spectral range from 4000 to 600 cm$^{-1}$ averaging 128 scans and a resolution of 2 cm$^{-1}$. Deionized water was used as background spectrum. All measurements were performed at room temperature. After each measurement, the E/P-co polymer layer was regenerated using a mixture of MeOH and water (1:10, *v/v*). In order to remove residual MeOH, pure deionized water was flushed through the BioATR cell II.

Data acquisition and processing was performed using the software OPUS and Essential FTIR, respectively. The obtained spectra were manually baseline corrected and evaluated using the peak area of the respective analyte.

## 3. Results and Discussion

### 3.1. Analysis of VOCs

Figure 2 shows an exemplary MIR spectrum, i.e., fingerprint region, of a mixture of nine VOCs (i.e., EB, TCE, TeCE, NAPH, TOL, BENZ, and the three xylene isomers) recorded using ATR-FTIR spectroscopy and a F-DLC and E/P-co polymer coated Si wafer. Aqueous solutions contained each analyte with a concentration of 25 ppm. All analytes were simultaneously investigated in deionized water. This spectrum was recorded after an enrichment time, i.e., diffusion of analyte molecules into the hydrophobic E/P-co polymer membrane, of 25 min. Due to substance specific C-H out-of-plane or C-Cl stretching vibrations, IR absorption peaks of the respective VOC occur. The IR signals used for further evaluation are labeled for clarity. Since the peaks are clearly separated, multivariate evaluation methods are not necessary, instead, the peak area was evaluated. The DLC and E/P-co layer showed no influence on the measurement performance. The MeOH concentration of 1% (*v/v*) do not influence the peak positions of the analytes.

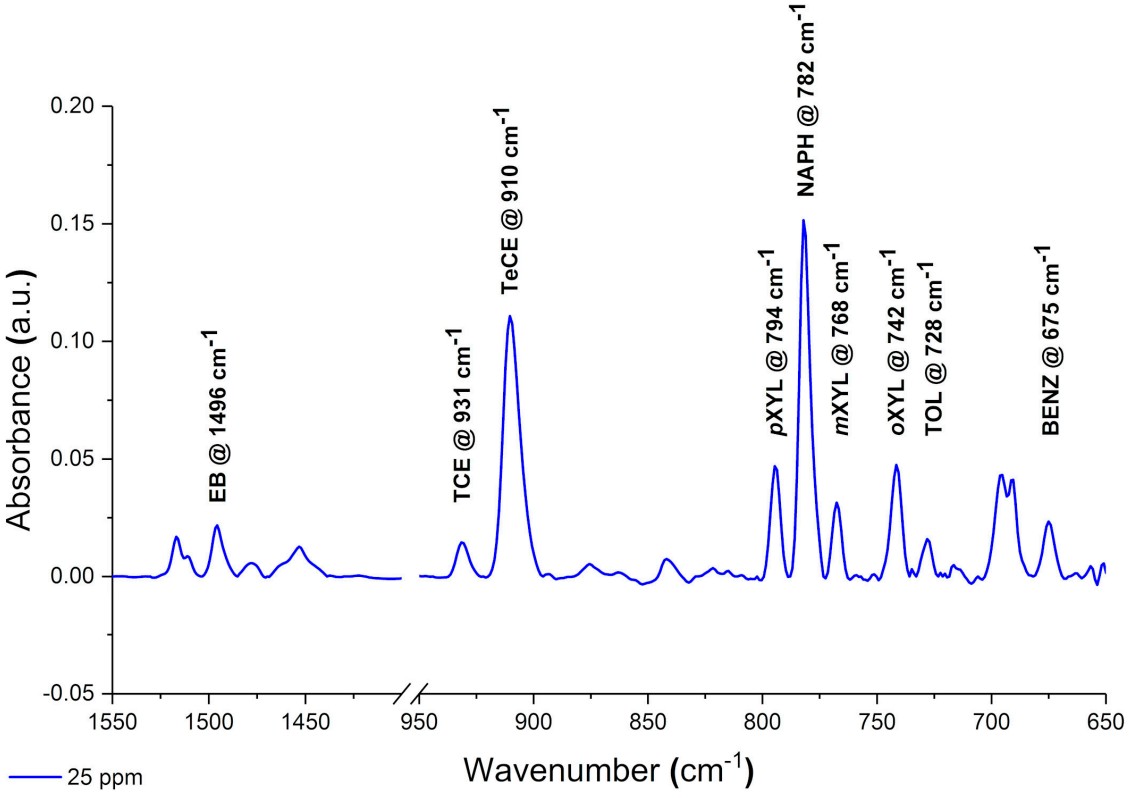

**Figure 2.** Exemplary ATR-FTIR spectrum of nine simultaneously detected VOCs in deionized water recorded with a fluorine-terminated DLC-coated Si wafer and an E/P-co polymer coating after 25 min of diffusion/enrichment. The IR absorption features do not overlap; therefore, univariate analysis was applied.

### 3.2. Diffusion of VOCs into Polymeric Membrane

Figure 3 shows the enrichment of TeCE at six different concentrations (flow rate: 8.7 mL/min) into the hydrophobic E/P-co polymer layer., i.e., the evaluated peak area at the specific wavenumber, as a function of enrichment time. 1, 5, 10, 15, 20, and 25 ppm are represented by black, red, blue, green, pink, and orange, respectively. The partitioning process was fitted using an exponential function. The parameters are summarized in Table 1. Depending on the concentration and type of analyte, and therefore, the partition coefficient between water and the E/P-co polymer membrane, time until equilibrium is achieved varies. However, at the latest within the first 5 min, 90% of diffusion

equilibrium ($t_{90}$), i.e., constant peak area, of all analytes was achieved. The $t_{90}$-values from this work and other research studies are summarized in Table 2 with the investigated analytes and the utilized polymer layer.

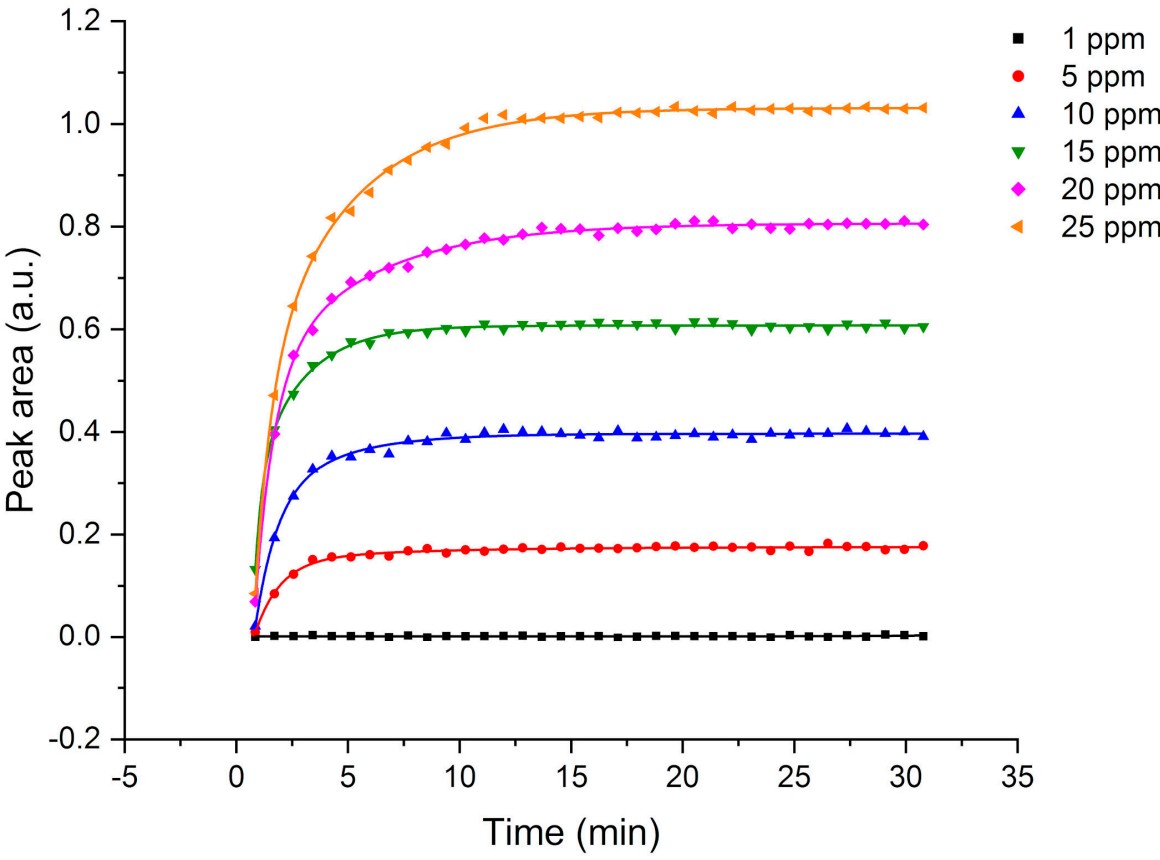

**Figure 3.** Diffusion curves of TeCE with the concentrations 1 (black), 5 (red), 10 (blue), 15 (green), 20 (pink), and 25 ppm (orange) into E/P-co polymer coated F-DLC Si wafer. The flow rate was set to 8.7 mL/min. The data points were fitted using an exponential fit function.

**Table 1.** Coefficients for exponential fitting the enrichment functions and $t_{90}$-value of TeCE into E/P-co polymer membrane at the concentrations 1, 5, 10, 15, 20, and 25 ppm [a].

| Concentration (ppm) | $A_1$ | $t_1$ | $A_2$ | $t_2$ | $y_0$ | $r^2$ | $t_{90}$ (min) |
|---|---|---|---|---|---|---|---|
| 1 | $7.94 \times 10^{-8}$ | −3.1012 | $9.95 \times 10^{-5}$ | $1.47 \times 10^{113}$ | 0.00155 | 0.06868 | - |
| 5 | −3.007 | 1.12151 | −0.03009 | −6.41271 | 0.1756 | 0.98684 | 4.95 |
| 10 | −0.59011 | 0.94089 | −0.18119 | −3.15137 | 0.39664 | 0.99169 | 5.02 |
| 15 | −2.6774 | 0.32479 | −0.4226 | −2.13323 | 0.60739 | 0.99646 | 4.14 |
| 20 | −1.21212 | 0.84059 | −0.35848 | −4.71451 | 0.80619 | 0.99799 | 7.05 |
| 25 | −1.36697 | 0.69688 | −0.67895 | −3.92399 | 1.03101 | 0.99829 | 7.40 |
| [a] **Exponential fit function** | | | $y = A_1 \cdot \exp\left(-\frac{x}{t_1}\right) + A_2 \cdot \exp\left(-\frac{x}{t_2}\right) + y_0$ | | | | |

**Table 2.** Overview of response times ($t_{90}$-values) for detection of VOCs in water with the respective utilized polymer membrane.

| Analytes | Polymer | $t_{90}$-Value (min) | Ref. |
|---|---|---|---|
| BENZ, TOL, XYL | E/P-co | 18 | [1] |
| BENZ, TOL, XYL | E/P-co | 18 | [41] |
| TeCE, TCE | E/P-co | 27 | [52] |
| XYL, TeCE | E/P-co | 14–27 | [53] |
| TOL, TCE | PTFE | 25 | [31] |
| TeCE, TCE | LDPE/PIB | 23 | [54] |
| TOL, *p/m*XYL NAPH | PIB | 5–10 | [55] |
| BENZ, TOL, XYL, EB, NAPH | PIB | 36 | [36] |
| EB, TCE, TeCE, XYL, NAPH, TOL, BENZ | E/P-co | 5–10 | This work |

*3.3. Calibration Functions*

The enrichment curves of the respective analytes were used for establishing calibration functions either via the equilibrium or kinetic method. Using the equilibrium method, the peak area at steady-state conditions of ten repetitive measurements of a concentration range from 1 to 25 ppm was evaluated. TOL and *o*XYL could not be measured within 1 ppm, therefore, a concentration of 30 ppm was added to the calibration set. In the case of the kinetic method, the slope of the tangent applied to the linear range in the first few minutes is plotted vs. the concentration, i.e., the first two to four measurements. Within this study, the slope of tangent was calculated from the first derivative of the linear region. Figures 4 and 5 show the established calibration functions using the equilibrium and kinetic method, respectively. EB is displayed in black, TCE in red, TeCE in dark blue, *p*XYL in dark green, NAPH in pink, *m*XYL in orange, *o*XYL in brown, TOL in light blue, BENZ in light green. It is clearly apparent, that the kinetic method leads to less accurate results with increased error bars. However, this evaluation method is used to obtain rapid results, i.e., 2 to 3 min. This is important for a rapid detection of pollutants if polluted wastewater is disposed in industrial plants, or leakages in pipelines or underground fuel storage tanks occur. Therefore, contaminated areas can be detected quickly. Table 3 summarizes the linear fit functions with the goodness of fit ($r^2$-value) for each investigated analyte with its respective peak area evaluated using the equilibrium method. In Table 4, the linear fit functions and $r^2$-values are listed using the kinetic method. The peak area remained the same. The obtained $r^2$-values of both evaluation methods are >0.98 to 0.999 for all analytes, except for TOL and BENZ resulting from a lower partitioning into the hydrophobic membrane compared to other VOCs. However, using the equilibrium method, slightly better $r^2$-values were obtained.

**Table 3.** Overview of the calibration functions established from evaluating the peak area at equilibrium conditions with the evaluated wavenumber, linear fit function, and $r^2$-value of the respective analyte.

| VOC | Wavenumber (cm$^{-1}$) | Linear Fit | $r^2$ | Linear Range |
|---|---|---|---|---|
| **EB** | 1496 | $0.00654x - 0.00611$ | 0.99984 | 1–25 |
| **TCE** | 931 | $0.00346x - 0.00327$ | 0.99892 | 1–25 |
| **TeCE** | 910 | $0.04282x - 0.04103$ | 0.99977 | 1–25 |
| ***p*XYL** | 794 | $0.0096x + 0.00085$ | 0.99944 | 1–25 |
| **NAPH** | 782 | $0.04083x - 0.01879$ | 0.99968 | 1–25 |
| ***m*XYL** | 768 | $0.00628x + 0.00263$ | 0.99848 | 1–25 |
| ***o*XYL** | 742 | $0.01576x - 0.07358$ | 0.99761 | 5–30 |
| **TOL** | 728 | $0.00431x - 0.02675$ | 0.97877 | 5–30 |
| **BENZ** | 675 | $0.00471x - 0.00389$ | 0.99273 | 1–25 |

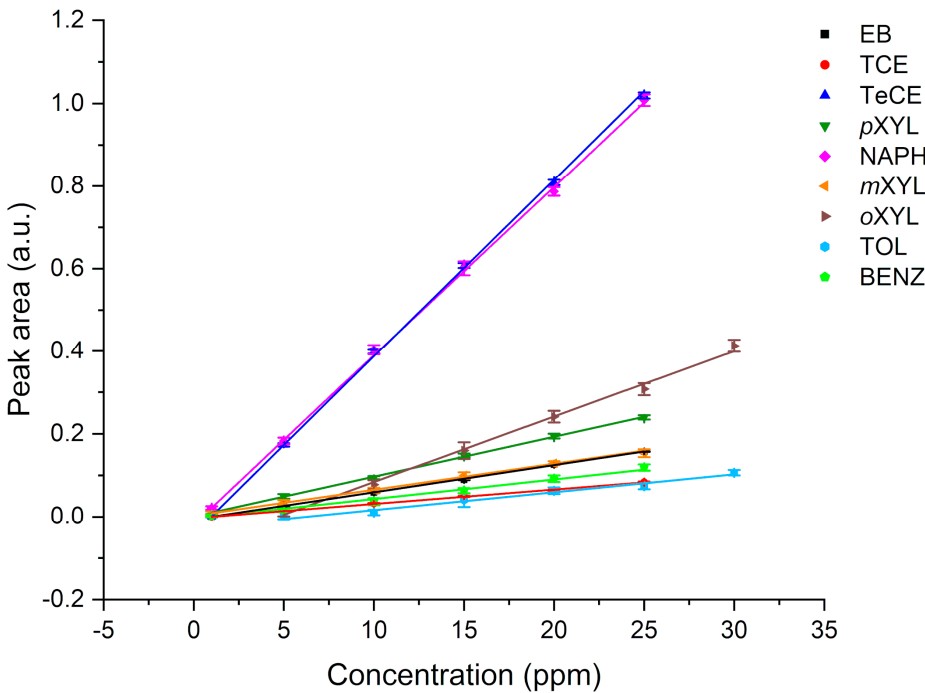

**Figure 4.** Established calibration functions using the equilibrium method. Black: EB, red: TCE, dark blue: TeCE, dark green: *p*XYL, pink: NAPH, orange: *m*XYL, brown: *o*XYL, light blue: TOL, and light green: BENZ. The concentrations 1, 5, 10, 15, 20, and 25 ppm were used. In the case of TOL, 5 to 30 ppm was measured. Each data point represents 10 repetitive measurements after achieving steady-state conditions.

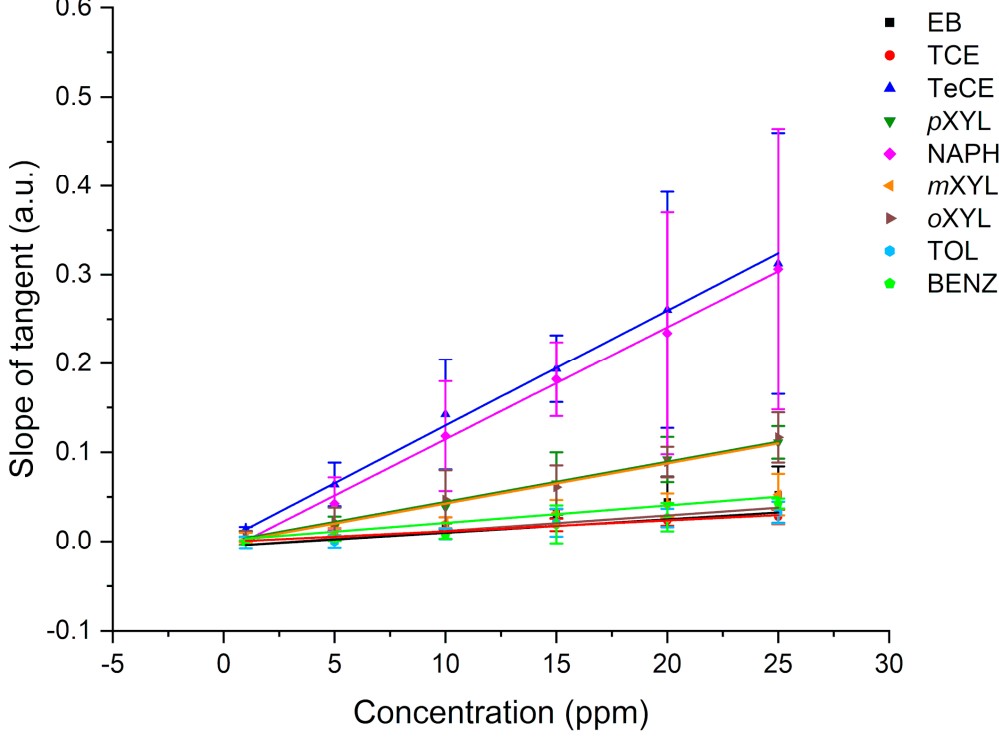

**Figure 5.** Calibrations functions in the concentration range from 1 to 25 ppm obtained using the kinetic method, i.e., evaluating the slope of the tangent of the linear range of enrichment curve. Black: EB, red: TCE, dark blue: TeCE, dark green: *p*XYL, pink: NAPH, orange: *m*XYL, brown: *o*XYL, light blue: TOL, and light green: BENZ.

**Table 4.** Linear fit functions and $r^2$-values of the respective VOC. The calibration functions were obtained using the kinetic method, i.e., evaluating the slope of the tangent in the linear range of enrichment curve. The linear concentration was from 1 ppm to 25 ppm for each VOC.

| VOC | Linear Fit | $r^2$ |
|---|---|---|
| **EB** | $0.0021x - 0.00108$ | 0.99252 |
| **TCE** | $0.0012x - 0.000446$ | 0.99274 |
| **TeCE** | $0.01293x + 0.000752$ | 0.99869 |
| *p***XYL** | $0.00438x - 0.0024$ | 0.98308 |
| **NAPH** | $0.0126x - 0.01148$ | 0.99643 |
| *m***XYL** | $0.00196x + 0.00132$ | 0.98105 |
| *o***XYL** | $0.00449x - 0.000419$ | 0.99716 |
| **TOL** | $0.00152x - 0.00531$ | 0.96863 |
| **BENZ** | $0.00174x - 0.00549$ | 0.9371 |

The limit of detection (LOD) and limit of quantification (LOQ) was calculated using the 3σ- and 10σ-criteria, thus, 3-times and 10-times the standard deviation of blank measurement, i.e., deionized water. Table 5 shows an overview of established detection and quantification limits using either the equilibrium or kinetic method. Through comparison of applied evaluation methods, the LODs and LOQs are slightly higher by the use of the kinetic method, but this is sufficient for fast real-time detection of pollutants in landfill or industrial effluents, and sewages. In summary, the obtained LODs are in the low ppb (50 ppb for TeCE) to low ppm concentration range showing the feasibility of F-DLC coated Si wafers and application of E/P-co polymer membrane as enrichment layer for multiple VOCs. The sensitivity is mainly influenced by the partitioning behavior, from the aqueous solution into the E/P-co polymer layer, thus, *m*XYL, TOL, and BENZ revealed lower detection limits. The detection limits obtained for nine simultaneously detected VOCs in water are comparable to other research teams [1,41,56,57]. Alternatively, silver halide fibers have already shown promising results as its use as sensing element [16,24,27,52,58]. Application of a F-terminated DLC layer combined with a silver halide fiber is feasible. Due to an increased number of total internal reflections a further decrease in LOD is expected.

**Table 5.** Comparison of detection (LOD) and quantification limit (LOQ) obtained using the equilibrium method and the kinetic method.

| VOC | Equilibrium Method | | Kinetic Method | |
|---|---|---|---|---|
| | LOD (ppm) | LOQ (ppm) | LOD (ppm) | LOQ (ppm) |
| **EB** | $0.12 \pm 0.0014$ | $0.41 \pm 0.0046$ | $0.39 \pm 0.091$ | $1.3 \pm 0.30$ |
| **TCE** | $0.37 \pm 0.0056$ | $1.2 \pm 0.016$ | $1.1 \pm 0.13$ | $3.7 \pm 0.24$ |
| **TeCE** | $0.050 \pm 0.0046$ | $0.17 \pm 0.0015$ | $0.16 \pm 0.031$ | $0.54 \pm 0.10$ |
| *p***XYL** | $0.32 \pm 0.0029$ | $1.1 \pm 0.0097$ | $0.72 \pm 0.10$ | $2.4 \pm 0.34$ |
| **NAPH** | $0.22 \pm 0.0015$ | $0.73 \pm 0.0049$ | $0.76 \pm 0.15$ | $2.5 \pm 0.49$ |
| *m***XYL** | $1.1 \pm 0.021$ | $3.6 \pm 0.070$ | $3.8 \pm 0.92$ | $12.6 \pm 3.05$ |
| *o***XYL** | $0.54 \pm 0.030$ | $1.8 \pm 0.10$ | $1.5 \pm 0.24$ | $4.8 \pm 0.78$ |
| **TOL** | $3.6 \pm 0.23$ | $12.1 \pm 0.77$ | $9.5 \pm 2.4$ | $31.6 \pm 8.15$ |
| **BENZ** | $0.97 \pm 0.031$ | $3.2 \pm 0.10$ | $3.3 \pm 0.78$ | $10.9 \pm 2.59$ |

*3.4. Recovery of the Polymeric Membrane*

After each measurement, the polymer membrane was recovered using a mixture of MeOH and deionized water 1:10 (*v/v*) following a flushing of solely deionized water. Since the sensor has to be quickly ready for further operation, the maximum flow rate of 12 mL/min was applied. Figure 6 exemplarily shows the regeneration process (decreasing peak area vs. time) after enriching NAPH in a concentration of 25 ppm. The coefficients for the exponential fit of the regeneration curve are given in Table 6. As with the enrichment process, the regeneration is diffusion-controlled. Therefore,

depending on the analyte and its concentration, time until complete removal varies. Regeneration times, i.e., until the initial peak area is achieved, reach from 10 to 30 min. In Figure 7, the entire measurement cycle for four analytes with a concentration of 15 ppm (a) TCE, (b) TeCE) and 25 ppm (c) NAPH, (d) *p*XYL) is illustrated. Black represents the diffusion into the E/P-co polymer, red is the polymer recovery using MeOH/water, and blue is associated with the flushing of water to remove MeOH residuals. The overall measurement time is around 1 to 1.5 h, afterwards, new measurements can be performed.

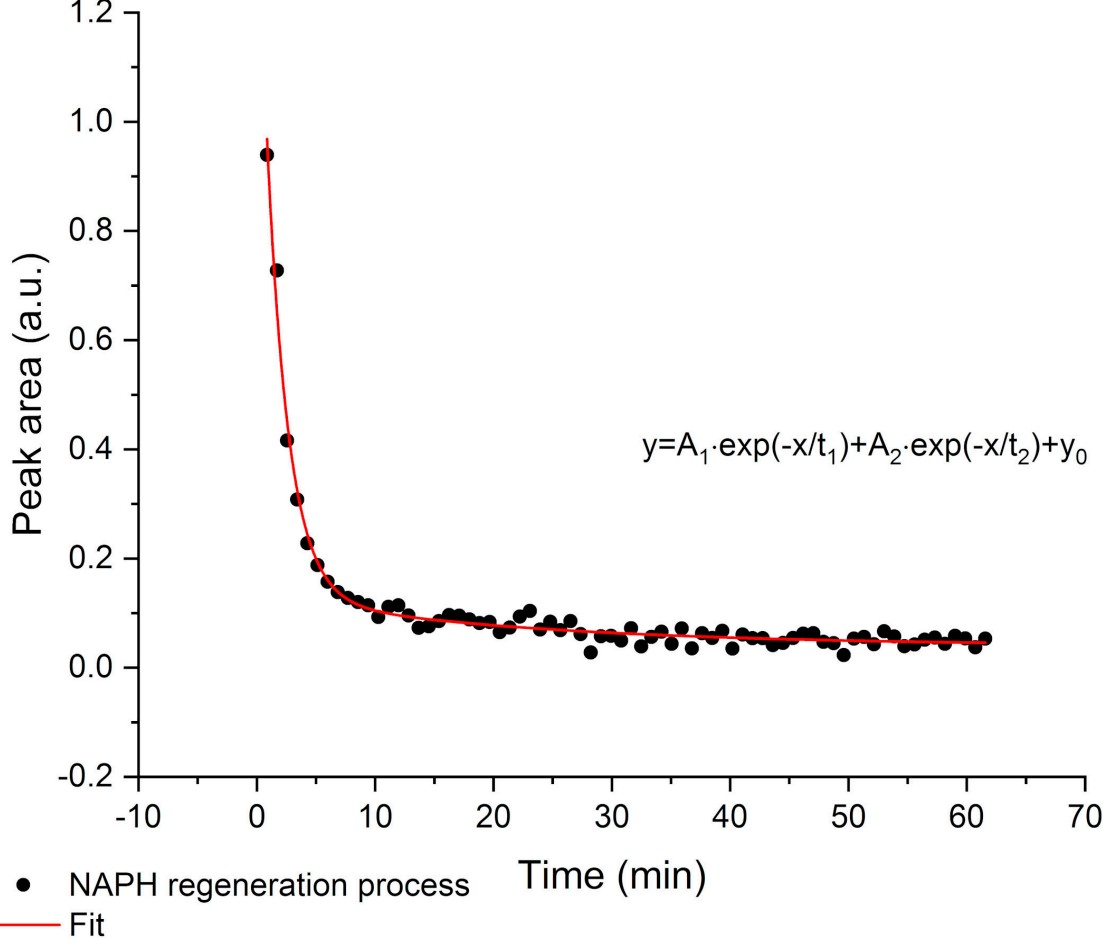

**Figure 6.** Regeneration of the E/P-co polymer layer after enrichment of 25 ppm NAPH. A MeOH/water mixture (1:10, *v/v*) was used for recovering the polymer membrane. Subsequently, deionized water was flushed through to remove residual MeOH before starting the next measurement. For the entire regeneration process, the maximum flow rate of 12 mL/min was applied. An exponential fit was applied to the experimental data points.

**Table 6.** Exponential fit function for the regeneration process with MeOH/water and subsequently water after enriching 25 ppm NAPH [a].

|  | $A_1$ | $t_1$ | $A_2$ | $t_2$ | $y_0$ | $r^2$ |
|---|---|---|---|---|---|---|
| **Regeneration process** | 1.34466 | 1.81229 | 0.09328 | 22.11773 | 0.03991 | 0.98679 |
| [a] **Exponential fit function** | $y = A_1 \cdot \exp\left(-\frac{x}{t_1}\right) + A_2 \cdot \exp\left(-\frac{x}{t_2}\right) + y_0$ | | | | | |

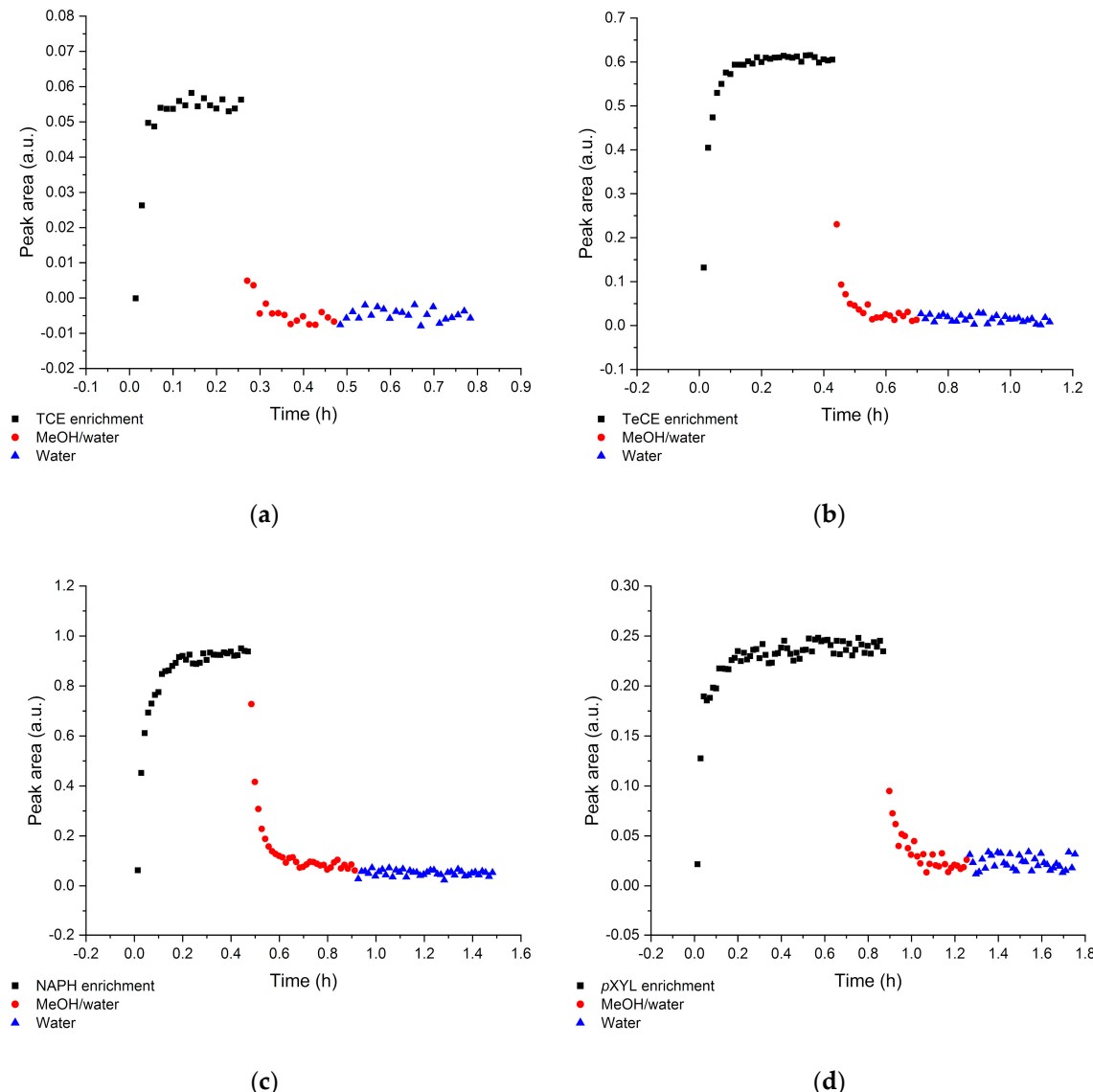

**Figure 7.** Entire measurement cycle exemplarily shown for (**a**) TCE (15 ppm), (**b**) TeCE (15 ppm), (**c**) NAPH (25 ppm), and (**d**) *p*XYL (25 ppm). Black: enrichment of analyte into E/P-co polymer layer, red: regeneration of polymer membrane using MeOH/water, and blue: flushing with water to remove residual MeOH.

The E/P-co polymer resists to multiple washing processes with MeOH showing a good attachment to the hydrophobic DLC layer. The long-term stability over several months was ensured during this study. This is very important for future applications in environmental analysis if silver halide fibers are employed directly in-field since they are sensitive to UV radiation and chloride ions if detection in a seawater matrix is performed. The hydrophobic coating acts both as enrichment membrane and protection for the optical fiber. The DLC layer would support the protective function. Within this study, a good adherence to the waveguide surface was observed ensuring long-term stability of silver halide fibers. Regarding the achieved selectivity and sensitivity and the entire measurement time, the Si wafer with F-terminated DLC and E/P-co polymer membrane presented within this study showed a good sensing performance with feasibility for environmental monitoring.

### 3.5. Varying Flow Velocities

Finally, the influence of different speed velocities of the peristaltic pump was investigated. Enrichment curves at four different flow rates (black: 5.9, red: 6.7, blue: 8.7, and green: 9.5 mL/min) were recorded. This is exemplary shown for 15 ppm EB in Figure 8. A summary of the exponential fit functions and $t_{90}$-values are given in Table 7. As expected, the $t_{90}$ time is higher when applying higher flow rates, however, this is not a significant difference. The results obtained are still in an acceptable time-range, i.e., 4 to 5 min. Furthermore, a minimum increase in peak area was observed when applying higher flow rates resulting from an increased mass transfer due to an increased contact with analytes molecules. Therefore, the equilibrium concentrations at the four respective flow rates were exemplarily plotted in Figure 9 (a: TCE, b: *m*XYL, 15 ppm each) revealing no significant difference between the four flow rates. This behavior was already observed previously by Yang et al. [31]. All VOCs above a vapor pressure of 1 Torr showed no significant influence of the flow rate on the equilibrium concentrations. However, if the flow rate is set too high, the polymer membrane may rupture causing leakages.

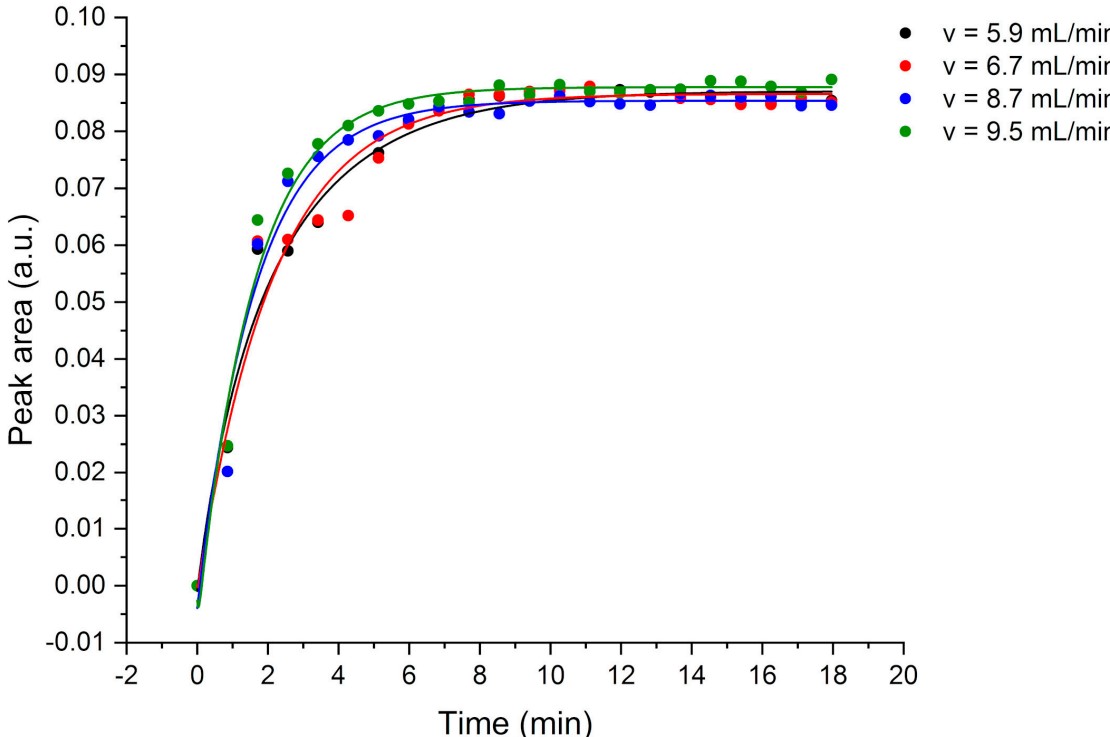

**Figure 8.** Comparison of enrichment curves at different flow rates exemplarily shown for EB at a concentration of 15 ppm. Black: 5.9 mL/min, red: 6.7 mL/min, blue: 8.7 mL/min, green: 9.5 mL/min.

**Table 7.** Overview of coefficients for exponential fits and $t_{90}$-value for enrichment functions of EB at 5.9, 6.7, 8.7, and 9.5 mL/min [a].

| Flow Rate (mL/min) | $A_1$ | $t_1$ | $A_2$ | $t_2$ | $y_0$ | $r^2$ | $t_{90}$ (min) |
|---|---|---|---|---|---|---|---|
| 5.9 | −0.01949 | 0.80129 | −0.0685 | 2.69548 | 0.08707 | 0.97421 | 5.57 |
| 6.7 | 0.00131 | 0.01401 | −0.08794 | 2.13971 | 0.08658 | 0.96748 | 4.96 |
| 8.7 | −0.04462 | 1.64621 | −0.04467 | 1.64621 | 0.08538 | 0.97664 | 3.86 |
| 9.5 | −0.09527 | 1.58738 | 0.00488 | 0.03228 | 0.08776 | 0.98329 | 3.78 |
| [a] **Exponential fit function** | | | $y = A_1 \cdot \exp\left(-\frac{x}{t_1}\right) + A_2 \cdot \exp\left(-\frac{x}{t_2}\right) + y_0$ | | | | |

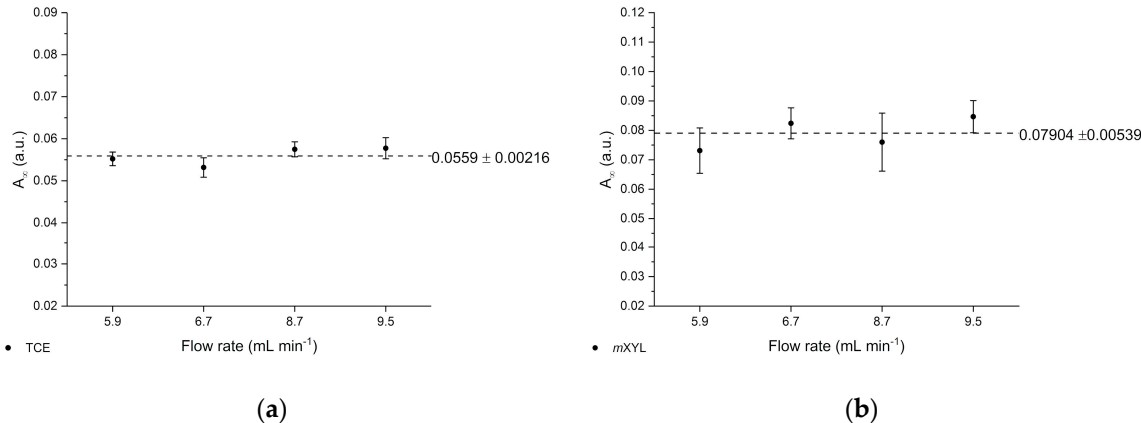

**Figure 9.** Comparison of equilibrium concentrations of TCE (**a**) and *m*XYL (**b**) each in a concentration of 15 ppm at four different flow rates: 5.9, 6.7, 8.7, and 9.5 mL/min. After achieving steady-state conditions, 10 repetitive measurements were performed. The dashed line represents the calculated mean value of the four flow rates.

## 4. Conclusions

This study shows the potential of employing a F-DLC coated Si wafer in combination with an E/P-co polymer as enrichment membrane for analysis of VOCs and pollutants in aqueous matrices via ATR-FTIR spectroscopy. The DLC coating was not influencing the IR absorption features, therefore, nine different VOCs were simultaneously detected, and the results showed a high sensitivity in the ppb to ppm concentration range. Detection is very fast due to direct response of the sensor, i.e., change in peak area, if the analyte reaches the polymer layer. Depending on the evaluation method, i.e., evaluating the concentration at steady-state conditions (equilibrium method) or the slope of the tangent of the enrichment curve (kinetic method), results are obtained within 5–10 min or 2–3 min, respectively. If a high accuracy is required, the equilibrium method should be applied. If a fast detection of pollutants is required, to detect leaking pipelines or underground fuel tanks the kinetic method is advantageous since accurate results are not necessary. Mainly the detection of pollutants is important. Hence, real-time monitoring at sewage disposal or near effluents of industries is enabled. The polymer can be reversibly recovered using a MeOH/water mixture within 10–30 min. The long-term stability of E/P-co polymer in combination with F-DLC was successfully maintained despite a variety of regeneration processes containing MeOH. Silver halide fibers used as sensing elements offers an increased number of internal reflections, and therefore, a higher sensitivity. If the Si wafer is replaced by a silver halide fiber in future environmental applications, the stability over a long period of time of E/P-co polymer is very important since the silver halide fiber decomposes when exposed to light or in contact with chloride ions contained in seawater matrix. Within this study, the combination of F-terminated DLC film with hydrophobic polymer layers providing a sufficient protection of the waveguide due to strong adhesion was shown. Future investigations therefore include, implementation of F-DLC coated silver halide fibers with real seawater matrices for long-term monitoring applications. In conclusion, the study showed promising results for the potential of DLC coatings in combination with a hydrophobic polymer as protection layer for silver halide fibers for future integration in sensing applications via ATR-FTIR spectroscopy for environmental analysis.

**Author Contributions:** Conceptualization, C.D., D.T., M.K., B.M.; methodology, C.D., D.T., A.M.; data curation, C.D.; writing—original draft preparation C.D.; writing—review and editing M.K., L.Ö., B.M.; supervision A.M., L.Ö., M.K., B.M.; funding acquisition L.Ö., M.K., B.M. All authors have read and agreed to the published version of the manuscript.

**Funding:** This research was funded by the Horizon 2020 Framework Program of the European Union within the MSCA RISE Project TROPSENSE, grant number 645758 and the Swedish Research Council (VR), project 621-2014-5959.

**Acknowledgments:** The authors gratefully acknowledge Joakim Andersson for technical support. JAVU AB, Sweden is thanked for sample coating.

**Conflicts of Interest:** The authors declare no conflict of interest.

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
