# Peer review of "Determination of Volatile Organic Compounds in Water by Attenuated Total Reflection Infrared Spectroscopy and Diamond-Like Carbon Coated Silicon Wafers"

_chemosensors, doi:10.3390/chemosensors8030075_

Round 1
Reviewer 1 Report
The paper entitled: "Determination of volatile organic compounds in water by attenuated total reflection infrared spectroscopy and diamond-like carbon-coated silicon wafers" submitted by Dettenrieder et al., presents the investigation results on the potential of F-terminated DLC-coated Si wafers in combination with a hydrophobic E/P-co polymer layer for pollution monitoring via ATR-FTIR spectroscopy. The paper is well-organized and well-written. The subject is worth to be investigated as well as the paper brings new light for measurement method for VOC detection in the ppb-ppm range. Moreover, the developed sensor responds very fast, which is a crucial parameter for environmental applications such as fast detection of leaks. The Figures are well-presented. Therefore I recommend accepting the paper as it is.
Author Response
Dear Sir or Madam,
Thank you very much for reviewing the article.
Kind regards,
Carina Dettenrieder
Reviewer 2 Report
The authors describe a method of determining volatile organic compounds in water using the attenuated total reflection (ATR) technique.
The authors go beyond the established state of the art by coating the silicon waveguide material with amorphous films of diamond-like carbon (DLC) with a fluorine surface passivation. This surface passivation enables a tight binding to a polymer film which rejects water molecules, allowing at the same time access of VOCs to the optical interaction zone. In this way VOC concentrations down into the ppb range could be achieved.
The paper is well organized and well written. The text is supported by high-quality pictures and informative figure captions.
I therefore recommend the paper for publication in Chemosensors.
Author Response
Dear Sir or Madam,
Thank you very much for the article review.
Kind regards,
Carina Dettenrieder
Reviewer 3 Report
The authors present determination of volatile organic compounds in water by attenuated total reflection infrared spectroscopy and diamond-like carbon coated silicon wafers. The idea is very well conceived and the paper is clearly presented and structured. However, there are some points that need to be carefully considered. Hence, I want to reconsider the paper publication based on the major revision of the comments provided in the next:
1- The introduction section is very general and extensive. Although it contains all the information a reader may be looking for, but providing a little details of the basics and strategical explaining the technical details would be suffice. Hence, I recommend to decrease the introduction length by omitting the general subjective information and provide technical details with recently published papers. Also, include the novelty of this work along with the most important results in the last paragraph of the introduction section.
2- All the abbreviations should be defined at the first instance of their usage in the manuscript.
3- Provide CAS numbers of all the chemicals, reagents and substrates used for sensor fabrication.
4- The schematic representation of the sensor layout is shown in Figure 1. It would be great if the authors share the real-time scanning Electron Microscope or Transmission Electron Microscopic cross-sectional images of the sensors?
5- For clarity, the graphs in Figures 4, 5 and 8 should be presented in a way that they are distinguishable in the black and white version of the manuscript.
6- I want the authors to briefly elaborate why there work is novel and different from the recently published papers in the field. Provide a comparative analysis between the sensors response in present study and recently published papers in a table for readers' understanding.
Author Response
Dear Sir or Madam,
Thank you for your valuable comments and suggestions to improve our manuscript. My responses are listed below point-by-point. Track changes were enabled and are highlighted to indicate the changes.
Best regards,
Carina Dettenrieder
1- The introduction section is very general and extensive. Although it contains all the information a reader may be looking for, but providing a little details of the basics and strategical explaining the technical details would be suffice. Hence, I recommend to decrease the introduction length by omitting the general subjective information and provide technical details with recently published papers. Also, include the novelty of this work along with the most important results in the last paragraph of the introduction section.
Thank you for this important comment. As suggested, details of the basics, i.e., equation (1) – Snell’s law – critical angle and equation (3) – evanescent field intensity, were added and the introduction has been shortened as suggested. The last paragraph of the introduction section was extended with the most important results. The novelty of the work includes application of DLC-coated waveguides with a hydrophobic fluorine surface termination combined with a hydrophobic polymer membrane in order to detect VOCs. Therefore, a tight adhesion of the polymer to the DLC layer is enabled. In order to detect very fast any leakages for example, a fast detection is inevitable for protection of human and animal health and the environment. This work showed the simultaneous detection of nine important VOCs with detection limits down to the ppb concentration range.
2- All the abbreviations should be defined at the first instance of their usage in the manuscript.
All abbreviations were checked. The abbreviation GaAs/AlGaAs was explained, “mercury-cadmium-telluride” was deleted in section 3.3. In the first section “introduction”, mid-infrared was abbreviated with MIR.
3- Provide CAS numbers of all the chemicals, reagents and substrates used for sensor fabrication.
The CAS numbers of all chemicals and reagents were added as requested.
4- The schematic representation of the sensor layout is shown in Figure 1. It would be great if the authors share the real-time scanning Electron Microscope or Transmission Electron Microscopic cross-sectional images of the sensors?
A real-time scanning Electron Microscope image of the cross-section was tried to obtain. However, the thickness of hydrophobic F-terminated DLC layer is 30 nm and the resolution is not sufficient to display the different layers. Therefore, we decided to show a schematic representation to give the reader an illustration of the utilized coated wafers.
5- For clarity, the graphs in Figures 4, 5 and 8 should be presented in a way that they are distinguishable in the black and white version of the manuscript.
Thank you very much for this comment. The Figures 4, 5 and 8 were edited to distinguish them in the black and white version of the manuscript, i.e., the shape of data points was changed accordingly to them of Figure 3.
6- I want the authors to briefly elaborate why there work is novel and different from the recently published papers in the field. Provide a comparative analysis between the sensors response in present study and recently published papers in a table for readers' understanding.
Recently published papers provide measurements and sensor developments based on polymer membranes in order to detect VOCs. Polymer membranes have to withstand the harsh environment and adhesion (strong binding) to the waveguide surface is inevitable. Particularly when using silver halide fibers as waveguide materials, protection is highly important to prevent its decomposition when the fiber is in contact with seawater. Therefore, sensor development beginning with silicon wafers is crucial for future application with optical fiber waveguides. Based on the possibility of surface termination, fluorine terminated DLC coatings provide a highly hydrophobic surface. Therefore, a strong binding to the hydrophobic polymer is enabled. To the best of our knowledge, this is the first time combining a F-terminated DLC coated waveguide with a hydrophobic polymer layer for detection of VOCs in the ppm/ppb concentration range. Furthermore, very fast responses due to strong increase in peak area after starting the measurement were achieved.
Round 2
Reviewer 3 Report
The authors have provided extensive response to the comments. I believe the manuscript is now ready for publication. Hence, I want to accept the manuscript publication in the present form.